# Smile Design and Treatment Planning—Conventional versus Digital—A Pilot Study

**DOI:** 10.3390/jpm13071028

**Published:** 2023-06-21

**Authors:** Andrea Maria Chisnoiu, Andreea Cristina Staicu, Andreea Kui, Radu Marcel Chisnoiu, Simona Iacob, Mirela Fluerașu, Smaranda Buduru

**Affiliations:** 1Department of Prosthodontics, Faculty of Dental Medicine, “Iuliu Hațieganu” University of Medicine and Pharmacy, 400349 Cluj-Napoca, Romania; maria.chisnoiu@umfcluj.ro (A.M.C.); andreea.cris.staicu@elearn.umfcluj.ro (A.C.S.); iacob.simona@umfcluj.ro (S.I.); fluerasu.mirela@umfcluj.ro (M.F.); dana.buduru@umfcluj.ro (S.B.); 2Department of Odontology, Endodontics and Oral Pathology, Faculty of Dental Medicine, “Iuliu Hațieganu” University of Medicine and Pharmacy, 400347 Cluj-Napoca, Romania; marcel.chisnoiu@umfcluj.ro

**Keywords:** aesthetic restorations, digital design, wax-up, mock-up, digital workflow

## Abstract

Introduction: Several methods are currently available for providing a preview of the prosthodontic treatment, including computer simulations, 3D models, wax-ups, and mock-ups. The aim of this study is to compare the aesthetic aspects and assessment of conventional versus digital prefigurative methods. Methods: The study included 5 patients and 3 observers, for each of whom a wax-up was made in both the conventional and digital techniques. The analog method, which implied a mock-up molding with a silicone matrix of the wax-up, was compared to a digital workflow, which consisted of a mock-up milling from a digital design. The patient’s clinical mock-ups were recorded with digital photographs and assessed for nine different criteria by three observers. Results: The analysis has shown a balanced assessment of the aesthetic criteria without any significant difference between the analog and digital prefigurative methods. Conclusions: Between the two wax-ups (conventional and digital), there were some variations in smile and dental criteria; however, the obtained data were very similar. When it comes to the smile criteria, the general average grades of the mock-ups conducted using the conventional method are slightly higher than the ones using the digital technique.

## 1. Introduction

The need for an aesthetic preview before starting definitive treatment is well documented [1,2]. There are several techniques and tools available for previewing the results of an aesthetic restoration. The traditional method uses the competencies and knowledge of the dental technician, who creates a diagnosis wax-up. The wax-up is a useful tool for selecting the ideal treatment, enhancing communication with the patient, and for provisional restoration construction. Subsequently, a good wax-up will dictate the final prosthesis fabrication [3]. Any diagnostic wax-up should be accurate, aesthetic, and feasible. Traditionally, the tooth surface is designed with a conventional wax-up technique as part of prosthodontic planning. The tooth contour and occlusion are modified by the addition of wax on the external tooth surface [4].

The modern, digital approach uses technologies such as computer-aided design and computer-aided manufacturing (CAD/CAM) or Digital Smile Design (DSD) [5]. With CAD/CAM technology, a 3D digital model of the patient’s teeth and surrounding structures can be created, allowing the dental professional to design and create a restoration that closely matches the patient’s natural teeth. In the same manner, Digital Smile Design allows the practitioner to make the wax-up digitally, following the pre-configurations of Smile Design software, and then 3D print it. The mock-up, being the representation of the wax-up in the patient’s arch, can therefore be derived either from the traditional cast or from the digital one [2]. Most prefigurative methods include the use of mock-ups or provisional restorations. These temporary restorations can be placed in the patient’s mouth to give them an idea of what their final restoration will look like. This allows the patient to provide feedback and adjust before the final restoration is created [6].

Studies have shown that using a mock-up can increase the patient’s trust in the treatment plan and thus his motivation throughout the treatment steps, as well as ensure predictable and satisfactory results [7,8]. Nonetheless, it is important to take into consideration the functionality of the restoration to obtain a durable, precise restoration as well as to avoid remakes. There are multiple possible objectives of an aesthetic restoration: treating a substance loss, a discoloration, or a shape anomaly. A smile design evaluation can also be performed through a direct mock-up by keeping the correct proportions and symmetries [9,10]. For all of these, prefigurative methods, wax-ups, and mock-ups have a very important role.

Studies have yet to agree unanimously on whether the conventional, analogical method or the fully digital one offer the best clinical outcomes [11,12]. Some objectives may be attained better by the fully digital methods, as they are more precise, easier, and quicker to use [12], while the analogic techniques might help the patient better understand and follow the treatment plan [13].

The digital wax-up has the advantages of not permanently altering the stone cast, quantifying the dental modifications, simplicity of execution, and the possibility of trying different treatments. Further, as the digital wax-up is performed using specialized software, more clinicians can provide a wax-up, even without artistic or technical abilities. However, the digital wax-up should at least exhibit a similar accuracy to conventional wax-up. Moreover, an increased interest is observed among dental professionals in using digital workflows in their practices [2,3,4,5,6,7,8,9,10].

Currently, the literature lacks information regarding conventional versus digital prosthetic wax-up, specifically with multiple practitioners involved.

The aim of this study is to evaluate and compare, using a validated questionnaire, the ability to preview the treatment plan and aesthetic dental restorations of the conventional analog system and the digital wax-up production methods from the point of view of dental professionals. The null hypothesis was that digital wax-up is superior in terms of aesthetics compared to the conventional previsualization method.

## 2. Materials and Methods

The pilot study included five patients and three observers.

The inclusion criteria for the voluntary participation of patients of both genders were: 20 to 25 years of age, in good general health, no active cavities or periodontal disease, at least four teeth with an indication of previous esthetic restorations with ceramic laminate veneers, no systemic diseases, and no allergies to the materials used.

For each of the patients, a wax-up has been made in both the conventional analog as well as the digital technique.

Initially, both facial and smile close-up photographs were captured using a single-lens reflex digital camera (NIKON D3500) with an AF-P 18 to 55 mm VR lens, lens mount NIKON F mount (with AF contact), and Nissin MF18 Macro Flash.

For each arch, an irreversible hydrocolloid impression (Alginate, GC America, IL, USA) was made. The impressions were poured with a type III dental stone (Gips Elite Model Blue Type III, Zhermack, Marl, Germany). These casts comprised the pre-treatment casts. All the casts were duplicated twice by silicone duplicating materials (Elite Double 22, Zhermack, Marl, Germany). One cast received a conventional wax-up and the other was scanned and used for the digital wax-up.

### 2.1. Conventional Wax-Up

After semi-adjustable articulator mounting of the pre-treatment casts (Bioart 7Plus, Bioart, Sao Carlos, Brazil), the conventional wax-up was completed by inlay wax (VITA Zahnfabrik, Bad Sackingen, Germany) addition on the buccal tooth surface. In some areas, the external tooth surface was modified by trimming. The wax-up aimed to rectify the defective tooth structure, establish natural and aesthetic tooth morphology, and achieve symmetry between the two sides [14]. All the conventional wax-ups were completed by an experienced dental technician.

### 2.2. Digital Wax-Up

The initial casts were scanned using the 3Shape Scanner, and STL images as well as initial smile evaluation photos were analyzed, superposed, and processed using the 3Shape Dental System Software (3Shape, Copenhagen, Denmark). On the virtual cast, the aim was to obtain an ideal tooth arrangement, emergence profile, symmetry, and aesthetics. Final digital wax-up models were printed using a Form 3D+ printer (Formlabs, Sommerville, MA, USA).

### 2.3. Mock-Up Procedure

C-silicone impressions (Putty and Light Speedex Coltene, Langenau, Germany) were taken of the wax-ups (both digital and conventional). The silicone guides were tried in the mouth, and any necessary corrections were made with a scalpel. Self-curing composite resin (Structur 3, shade A1, Voco, Cuxhaven, Germany) was applied to the guide and then inserted in the mouth. Adjustments were made after the final setting of the resin either by means of contouring high-speed burs in conjunction with water cooling, as in the case of conventional composites, or by filling defects with a flowable composite (Grandio Flow, Voco, Cuxhaven, Germany).

The patient’s clinical mock-ups were recorded with digital photographs (Figure 1) and evaluated by three experienced dentists (with more than 10 years of clinical experience). All images were taken at a real-life (1:1 ratio) size; hence, there was no magnification error, and they were displayed for analysis on the computer screen in the same chronology for each observer. The observers have responded independently to a set of questions, noted from 1 to 5 (Table 1). The aesthetic assessment questionnaire was created by the authors based on the research of Mocelin et al. [15]. Initial calibration training for the observers was given in order to standardize the research methodology. To ensure the validity and reliability of results, inter- and intra-examiner calibrations were conducted to assess the aesthetic result agreement between the examiners.

For each criterion, an average of all the obtained grades was registered. The statistical analysis was performed with the Statistical Package for Social Sciences, version 16.0 software (SPSS Inc.; Chicago, IL, USA). To test reliability and intra- and inter-evaluators agreement, Cohen’s kappa values were calculated. All values were below 0.8 and considered acceptable. The normality of the data distribution was analyzed using the Shapiro–Wilk test. The data obtained from the aesthetic evaluation were analyzed using the *t*-test, and the statistical significance was set at *p* < 0.005.

## 3. Results

Results have shown a balanced assessment of the aesthetic criteria the observers have graded between the analog and digital prefigurative methods. Figure 2 and Figure 3 show the rather even distribution of the grades given to each patient by the three observers (*p* > 0.005).

Even so, the mock-ups made using the digital technique were appreciated for being less precise and accurate regarding one particular criteria, the surface, scoring lower grades than the analog method. However, the shape of the mock-ups was considered superior when using the digital method versus the analog one.

For two of the patients, each of the criteria for the manual mock-up has a greater average than the results of the one derived from the 3D printed wax-up. This difference is found in the general assessment of the mock-ups, as the traditional method derived from analog wax-ups has a superior average to that made after the 3D-printed one. For three patients, the difference between the averages shows less discrepancy, and for certain criteria, the digital method did have a superior grade than the traditional one.

When it comes to the smile criteria, the general average grades of the mock-ups conducted by using the conventional method are higher than the ones using the digital technique, except for the antero-posterior position of the frontal teeth (Figure 4) (*p* > 0.005) (Table 2). On the other hand, the mock-ups using the digital technique have been perceived as offering better results when it comes to the dental aspects, such as the proportion of the teeth and the arch shape (Figure 4).

Regarding the general assessment of the cases, the mock-ups created using conventional analog methods have obtained slightly better results (Figure 5).

## 4. Discussion

The aim of this pilot study was to evaluate and compare, from the point of view of dental professionals and using an aesthetic questionnaire, the conventional analog system with the digital mock-up production method of clinical results prefiguration in dentistry. Overall, the results of this study were in accordance with the existing literature. There was no significant difference between the traditional system that uses a conventional wax-up made by the technician and the modern, digital method, so the null hypothesis was rejected. When analyzing the criteria taken into consideration in this study, both techniques have ensured better results for specific metrics, to the disadvantage of the other. On what concerns the proportion of the restorations and the arch shape in the mock-up stage, the digital method has shown a better average appreciation, while for the surface criteria, the same method has shown to be inferior to the conventional one. The results are similar to those of recent research published by Mocelin et al. [15], which investigated patient and dentist preference between conventional and digital diagnostic waxing and identified that the esthetics of the digital method were preferred in half of the cases and the conventional approach in the other half.

Many studies have shown how a result-prefiguration-based approach can allow the practitioner to provide patients with precise, reproducible, and predictable results, especially when it comes to aesthetics, since this particular kind of tool works with the psychology of patients, improving their understanding, motivation, and compliance towards the treatment [13,14,15,16]. It is highly recommended that practitioners have a clear treatment plan and desired outcome in mind before starting any case, particularly when altering the morphology of anterior teeth. As changes to the front teeth can significantly impact a patient’s smile and overall appearance, it is essential to have a clear vision of the desired outcome to ensure the best possible results, both functionally and aesthetically. Planning can also help the practitioner make the necessary adjustments throughout the treatment process to achieve the desired outcome. A diagnostic wax-up can be a useful tool in enhancing the predictability of treatment when changing the morphology of anterior teeth. As such, the use of a mock-up is critical to ensuring a correlation to the patient’s clinical situation, thus avoiding a result that appears optimal on the casts but does not correspond to the patient’s smile [16,17,18]. Wax-ups and mock-ups are reported to be widely used by periodontal surgeons as tools used to perform crown lengthening procedures to enable future restorations in specific cases [19].

The vast availability of materials when it comes to digital techniques, which are faster and especially less tiring or constraining, can make the practitioners aspire towards a standardization of prosthetic designs at the expense of the know-how of the dental technicians and the customization of each case they can provide. A study by Kollmuss et al. [20] also showed superior esthetic results for conventional waxing. The authors consider that dental professionals training is still incomplete as regards the use of digital technologies, a fact that may explain patient preference for conventional waxing.

The disadvantages of the conventional wax-up and diagnostic mockup include a significant investment of time, difficulties in making small changes for indecisive patients, and inaccuracies at the incisal edge and cervical margin. Digital techniques like digital scanners and 3D-printed mockups open the door for more efficiency.

It is important to know that dental software is becoming more and more performant to provide professionals with a greater diversity of previsualizations. The use of smile design software allows for interdisciplinary collaboration between practitioners, and this seems to improve the decision-making process, ultimately decreasing the amount of intra-oral adjustment, remake rates, and the overall dissatisfaction of the patient [21,22,23]. A fully digital workflow may be more reliable than conventional techniques and make use of planning software such as digital smile design. However, significant monetary and time investments are needed to use these techniques, which may not be feasible for a general dentist [22,23].

In the current research, three experienced observers evaluated the aesthetic appearance of previsualization techniques in dentistry, compared to previous studies that included only one evaluator, giving more precise results.

The current study presents several limitations. First, the reduced number of participants cannot give a precise image of the evaluation. Including more patients in the study would ensure more reliable results and should be considered. Another limitation is represented by the skills of the dental technician performing the mock-up. It is of high importance that the dental technician can deliver the best quality wax-up using CAD/CAM technologies or conventional additive ones based on experience, which will remain indispensable.

The results of the present study showed that the esthetic perception of the dentists is equilibrated when evaluating conventional and digital wax-up. Currently, there are various technological resources available on the dental market, and the authors consider that the main factor for successful dental aesthetic treatments is the learning curve and adaptation to new techniques.

## 5. Conclusions

Within the limitations of this study, it could be concluded that, in terms of aesthetics, the wax-up/mock-up procedures significantly improved the clinical situation. Between the two wax-ups (conventional and digital), there were some variations in smile and dental criteria; however, the obtained data were very similar. When it comes to the smile criteria, the general average grades of the mock-ups conducted using the conventional method are slightly higher than the ones using the digital technique. Although the implications of this inaccuracy are yet to be determined, it is likely that continuous technological improvements will enhance the digital wax-up outcome.

## Figures and Tables

**Figure 1 jpm-13-01028-f001:**
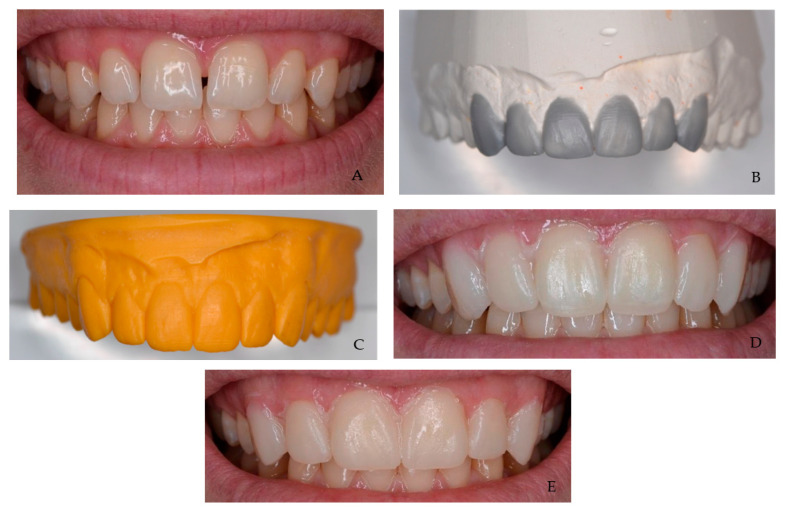
Photographs of analog and digital mock-ups. (**A**) Initial situation. (**B**) Conventional wax-up. (**C**) Digital wax-up. (**D**) Mock-up using the digital wax-up. (**E**) Mock-up using the conventional wax-up.

**Figure 2 jpm-13-01028-f002:**
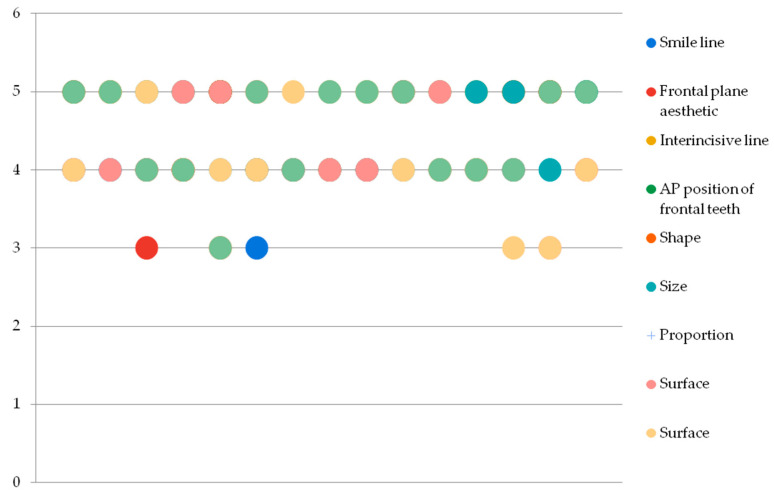
Grading distribution for the conventional method.

**Figure 3 jpm-13-01028-f003:**
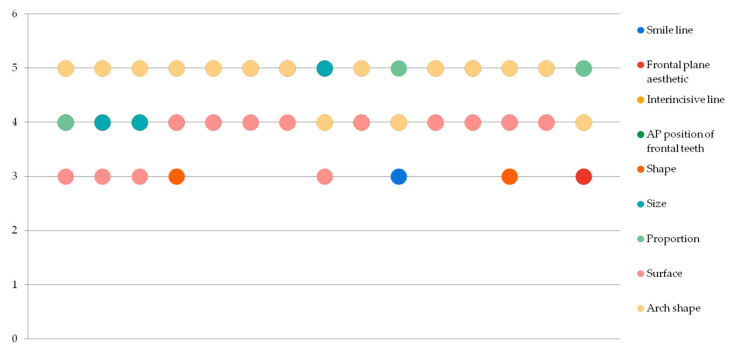
Grading distribution for the digital method.

**Figure 4 jpm-13-01028-f004:**
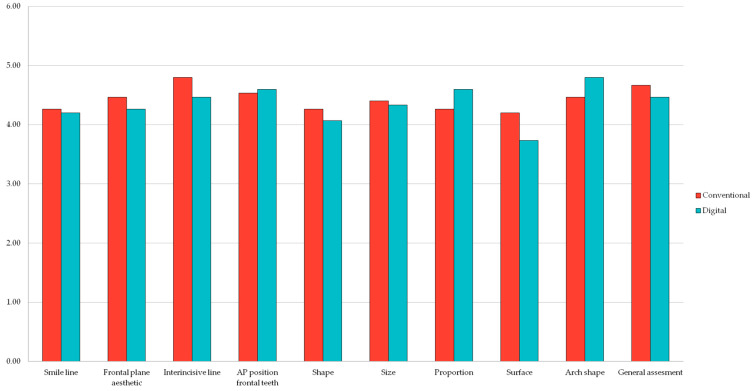
Comparison between conventional and digital methods regarding smile and dental criteria.

**Figure 5 jpm-13-01028-f005:**
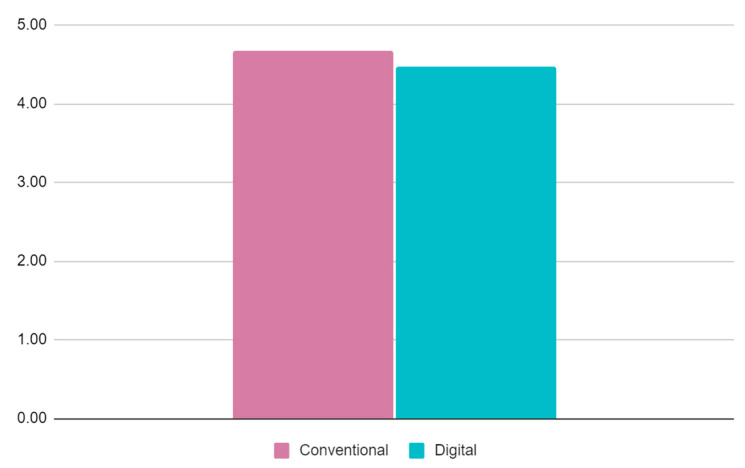
General assessment comparison between conventional and digital prefigurative methods.

**Table 1 jpm-13-01028-t001:** Aesthetic questionnaire.

			Conventional	Digital
Smile criteria	Frontal plane	Smile line	1 2 3 4 5	1 2 3 4 5
		Frontal plane aesthetic	1 2 3 4 5	1 2 3 4 5
	Sagittal plane	Interincisive plane	1 2 3 4 5	1 2 3 4 5
	Horizontal plane	Antero-posterior position of frontal teeth	1 2 3 4 5	1 2 3 4 5
Dental criteria		Shape	1 2 3 4 5	1 2 3 4 5
		Size	1 2 3 4 5	1 2 3 4 5
		Proportion	1 2 3 4 5	1 2 3 4 5
		Surface	1 2 3 4 5	1 2 3 4 5
		Arch shape	1 2 3 4 5	1 2 3 4 5
General assessment			1 2 3 4 5	1 2 3 4 5

Evaluation scale: 1—does not correspond; 2—low correspondence; 3—medium correspondence; 4—high correspondence; 5—completely corresponds.

**Table 2 jpm-13-01028-t002:** Mean values of evaluated criteria.

Criteria	Mean		*t*-Test *p* Value
	Conventional	Digital	0.40
Smile line	4.27	4.20	0.24
Frontal plane aesthetic	4.47	4.27	0.03
Interincisive line	4.80	4.47	0.38
AP position of frontal teeth	4.53	4.60	0.38
Shape	4.27	4.20	0.38
Size	4.40	4.33	0.38
Proportion	4.60	4.27	0.05
Surface	4.20	3.73	0.04
Arch shape	4.80	4.40	0.08
General assessment	4.67	4.47	0.18

## Data Availability

The data presented in this study are available upon request from the authors.

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
