# Peer review of "Smile Design and Treatment Planning—Conventional versus Digital—A Pilot Study"

_jpm, 2023, doi:10.3390/jpm13071028_

Round 1
Reviewer 1 Report
Abstract
· Headings in the summary text should be removed.
Introduction
· What is the "niche" point of the study in the literature? It should be clearly stated.
· More details should be specified in the purpose part of the study. For example, How are analog and digital methods compared?
Materials and methods
· What are the characteristics of the 5 patients and 3 observers mentioned in line 76? Please explain in detail.
· Details should be given about the three experienced dentists mentioned in lines 110-111. For example, how many years of experience do the relevant dentists have? Do they have any expertise in this matter?
· How was calibration achieved in dentists' evaluations?
· Express table 1 on lines 135-138 in the text. What does the 1 to 5 scoring in Table 1 mean? Please give details.
Results
· The colorings in figures 2 and 3 are not very clear. It should be redesigned.
· Combining figures 4 and 5 may be considered.
Discussion
· The results of this study have not been compared with the results of any concrete study. Please compare the results of this study in detail with related studies.
· The difference between this study and other studies was not explained. What gap in the literature does this study fill? Please specify in detail.
· No information was provided regarding the limitations of this study. Please specify.
· In short, the discussion part of the manuscript lacks basic data and evaluations. It is highly recommended that this section be rewritten in the light of this information.
There is no big problem in terms of language. However, the decimal apostrophe needs to be fixed as a period instead of a comma.
Author Response
Dear Reviewer,
Thank you for your valuable indications regarding the manuscript. We have made the changes and they are displayed in the table below.
|
Article section |
Reviewer indication |
Authors modification |
|
Introduction |
Headings in the summary text should be removed.
|
Headings have been removed |
|
|
What is the "niche" point of the study in the literature? It should be clearly stated. |
The „niche” point has been stated in the end of the introduction section. |
|
|
More details should be specified in the purpose part of the study. For example, How are analog and digital methods compared? |
Details have been given in the purpose part of the study. |
|
Materials and methods |
What are the characteristics of the 5 patients and 3 observers mentioned in line 76? Please explain in detail.
|
Characteristics of the patients and observers have been included. |
|
|
Details should be given about the three experienced dentists mentioned in lines 110-111. For example, how many years of experience do the relevant dentists have? Do they have any expertise in this matter?
|
Details about the three dentists have been added. |
|
|
How was calibration achieved in dentists' evaluations?
|
The calibration process has been explained |
|
Results |
Express table 1 on lines 135-138 in the text. What does the 1 to 5 scoring in Table 1 mean? Please give details.
|
The scoring has been explained |
|
|
The colorings in figures 2 and 3 are not very clear. It should be redesigned.
|
The colorings have been changed |
|
|
Combining figures 4 and 5 may be considered.
|
Figures 4 and 5 have been changed |
|
Discussion |
The results of this study have not been compared with the results of any concrete study. Please compare the results of this study in detail with related studies.
|
Comparison with other studies has been added |
|
|
The difference between this study and other studies was not explained. What gap in the literature does this study fill? Please specify in detail. |
Information added |
|
|
No information was provided regarding the limitations of this study. Please specify. |
Limitations of the study have been added |
|
|
In short, the discussion part of the manuscript lacks basic data and evaluations. It is highly recommended that this section be rewritten in the light of this information. |
The discussion section has been modified |
Reviewer 2 Report
The article “Smile design and treatment planning. Conventional versus digital – a pilot study” evaluate and compare the conventional, analog system with the digital wax-up production methods, to preview the treatment plan and aesthetic dental restorations
The article is interesting and helps in validating the mock-up procedure either in analogic and digital way.
One of the most important drawbacks is the number of cases and observers (quite low). The authors should support why and how these numbers were selected.
There is no null hypothesis and therefore it is not accepted or rejected in the discussion section. Please add.
Line 76: please describe the 5 cases at baseline (before the mockup):
1. age
2. gender
3. gingival smile
4. parallelism between incisal edge and lower lip (smileline)
5. etc.
Line 82, 84, place a comma (,) before Zhermack
Line 99: the authors cite symmetry, please add a proper reference like: PUBMED-ID: 24757696
Line 110: please describe the environment in which the observers evaluated the pictures: computer, laptop, scree, distance, randomized, etc.
Which type of statistical analysis was used?
Which test was used to test the normality distribution?
Where is the table? Just two values were related to (p>0,005) but without any scientific data.
Line 197:
In the sentence “analog methods have obtained slightly better results” please specify if it is a significant or not significant difference
Line 234:
The authors should also add a reference for direct restorations, not only for indirect ones. The reviewer suggests to add:
“Paolone G. (2017). Direct composites in anteriors: a matter of substrate. The international journal of esthetic dentistry, 12(4), 468–481.”
Please add further studies to develop based on the results of the current study
English language needs editing.
Author Response
Dear reviewer,
Thank you for your valuable indications regarding the manuscript. We have made the changes and they are displayed in the table below.
|
Reviewer suggestion |
Author response |
|
There is no null hypothesis and therefore it is not accepted or rejected in the discussion section. Please add. |
The null hypothesis was added |
|
Line 76: please describe the 5 cases at baseline (before the mockup): |
Supplementary data on cases at baseline was added. However, the full aesthetic analysis was not mentioned since the cases were previously analyzed and included in the study based on the previous need for aesthetic prosthodontic treatment including laminate veneers. – line 85, 86 |
|
Line 82, 84, place a comma (,) before Zhermack |
Comma has been added |
|
Line 99: the authors cite symmetry, please add a proper reference like: PUBMED-ID: 24757696 |
Reference added |
|
Line 110: please describe the environment in which the observers evaluated the pictures: computer, laptop, scree, distance, randomized, etc. |
Supplementary data has been added |
|
Which type of statistical analysis was used? |
Statistical analysis details have been added |
|
Which test was used to test the normality distribution? |
The test for normality distribution has been added |
|
Where is the table? Just two values were related to (p>0,05) but without any scientific data. |
A table has been added |
|
Line 197: |
Text has been modified |
|
Line 234: |
Reference has been added |
Reviewer 3 Report
This paper covers an interesting topic. Nevertheless some major improvements shall be performed before publication.
In particular:
Information on statistics is missing, please add.
An extensive english language editing should be performed.
Lines 40-1:
The authors wrote: “The modern, digital approach uses technologies such as computer-aided design and computer-aided manufacturing (CAD/CAM), or Digital Smile Design (DSD).”
Please add a reference for this sentence
Lines 49-52: add references please.
Line 79: please state if speedlight has been used
Please state if a facebow has been used
Line 109: describe type and brand of curing light and power in mW/cm2
After line 52 the authors should mention that direct mock-up could also be used in evaluating a possible treatment plan.
The following sentence could be added:
“A smile design evaluation can also be performed through a direct mock-up by keeping the correct proportions and symmetries.”
The following two references can be added:
1. Pubmed ID: 19655472
2. https://doi.org/10.3390/sym13050797
Please add limitations of current study:
skill of the dental technician
laboratory costs
etc.
This paper needs to be proofread by a native English speaker.
Author Response
Dear reviewer,
Thank you for your valuable indications regarding the manuscript. We have made the changes and they are displayed in the table below.
|
Reviewer suggestion |
Author response |
|
Information on statistics is missing, please add. |
Information on statistics has been added |
|
Lines 40-1: The authors wrote: “The modern, digital approach uses technologies such as computer-aided design and computer-aided manufacturing (CAD/CAM), or Digital Smile Design (DSD).” Please add a reference for this sentence |
Reference added |
|
Lines 49-52: add references please. |
Reference has been added |
|
Line 79: please state if speedlight has been used |
Speedlight information added |
|
Please state if a facebow has been used |
Facebow has been used as already mentioned in the text- line 99 |
|
Line 109: describe type and brand of curing light and power in mW/cm2 |
No curing light was used |
|
After line 52 the authors should mention that direct mock-up could also be used in evaluating a possible treatment plan. The following sentence could be added: “A smile design evaluation can also be performed through a direct mock-up by keeping the correct proportions and symmetries.”
|
Sentence added |
|
The following two references can be added: 1. Pubmed ID: 19655472 2. https://doi.org/10.3390/sym13050797
|
References have been added |
|
Please add limitations of current study: skill of the dental technician laboratory costs etc. |
Limitations of the study have been added. |
Round 2
Reviewer 1 Report
The recommended corrections have been made and it is appropriate to publish the manuscript as it is.
The recommended corrections have been made and it is appropriate to publish the manuscript as it is.
Reviewer 2 Report
Dear authors,
The paper is now suitable for publication.
Reviewer 3 Report
The authors have provided all the suggested improvements.